# Fatigue Damage Evaluation of Short Carbon Fiber Reinforced Plastics Based on Thermoelastic Temperature Change and Second Harmonic Components of Thermal Signal

**DOI:** 10.3390/ma14174941

**Published:** 2021-08-30

**Authors:** Daiki Shiozawa, Takahide Sakagami, Yu Nakamura, Takato Tamashiro, Shinichi Nonaka, Kenichi Hamada, Tomoaki Shinchi

**Affiliations:** 1Department of Mechanical Engineering, Kobe University, Kobe 657-8501, Japan; sakagami@mech.kobe-u.ac.jp (T.S.); 169t348t@stu.kobe-u.ac.jp (Y.N.); 140t348t@stu.kobe-u.ac.jp (T.T.); 2DIC Corporation, Tokyo 103-8233, Japan; shinichi-nonaka@mb.dic.co.jp (S.N.); kenichi-hamada@mb.dic.co.jp (K.H.); tomoaki-shinchi@mb.dic.co.jp (T.S.)

**Keywords:** nondestructive evaluation, thermoelastic stress analysis, phase analysis, infrared camera, short carbon fiber reinforced plastics

## Abstract

Short fiber reinforced plastics (SFRPs) have excellent moldability and productivity compared to continuous fiber composites. In this study, thermoelastic stress analysis (TSA) was applied to detect delamination defects in short carbon fiber reinforced plastics (SCFRPs). The thermoelastic temperature change Δ*T*_E_, phase of thermal signal *θ*_E_, and second harmonic temperature component Δ*T*_D_ were measured. In the fatigue test of SCFRP, it was confirmed that changes in Δ*T*_E_, *θ*_E_, and Δ*T*_D_ appeared in the damaged regions. A staircase-like stress level test for a SCFRP specimen was conducted to investigate the generation mechanism of the Δ*T*_D_. The distortion of the temperature change appeared at the maximum tension stress of the sinusoidal load—and when the stress level decreased, the temperature change returned to the original sinusoidal waveform. Δ*T*_D_ changed according to the change in the maximum stress during the staircase-like stress level test, and a large value of Δ*T*_D_ was observed in the final ruptured region. A distortion of the temperature change and Δ*T*_D_ was considered to be caused by the change in stress sharing condition between the fiber and resin due to delamination damage. Therefore, Δ*T*_D_ can be applied to the detection of delamination defects and the evaluation of damage propagation.

## 1. Introduction

Short fiber reinforced plastics (SFRPs) have received much attention due to their excellent moldability and productivity, as compared to continuous fiber composites. The mechanical properties of an SFRP depend on the fiber length and orientation distribution. A comprehensive literature review of SFRPs was published by Mortazavian and Fatemi [1]. The effects of loading conditions, microstructure, and environmental factors on the lifetime of SFRPs were investigated. Belmonte et al. [2,3] presented the influence of the fiber volume fraction on the damage mechanism in a short glass fiber. An effective nondestructive technique is required to detect and measure the various types of damage during fatigue fracture. Fragoudakis [4] discussed the effect of the fiber orientation around geometric discontinuities in glass fiber reinforced plastic (GFRP) and presented an important topic for the manufacturing and design against failure of GFRP laminated structures. Nondestructive evaluation techniques using infrared thermography (thermographic NDT) have been effectively employed for the detection of delamination in FRPs. The thermographic NDT technique based on the thermal insulation effect of the delamination defect was applied to the NDT of composite materials. Avdelidis et al. [5,6] reviewed NDT techniques using a transient temperature distribution for CFRP. Chatterjee et al. [7] compared the defect detection performance of transient thermography, pulse lock-in, and frequency-modulated thermography. Maldague et al. [8] developed pulse-phase infrared thermography for composite materials.

Thermoelastic stress analysis (TSA) is a well-known, non-contact, full-field technique that provides stress maps of a component subjected to dynamic loading [9,10,11,12,13]. Thermoelasticity for orthotropic materials has been studied by many researchers, and the TSA technique has been employed as a powerful tool for evaluating the impact or fatigue damage in composite materials and structures [14,15,16,17,18]. Krstulovic-Opara et al. [19] applied a combination of TSA and pulse heating thermography to detect damage, including fiber fractures and delamination. Emery and Dulieu-Barton [20] applied TSA to fatigue damage evaluation in laminated glass fiber epoxy materials, and demonstrated that TSA could analyze complicated fatigue damages, such as fiber breakage, matrix cracking, and delamination. Uenoya and Fujii [21] developed a thermoelastic damage analysis (TDA) for early damage detection in plain-woven CFRP.

A thermal signal analysis approach was developed to assess the different harmonic components of the thermal signal related to both the thermoelastic and dissipated heat sources. Palumbo et al. [22] showed that the amplitude of the second harmonics of the thermal signal of GFRP laminates was representative of the intrinsic dissipations, and it can be used to detect and monitor the damage. De Finis et al. [23,24] showed that the second harmonic of the thermal signal and damage state parameter of quasi-isotropic CFRP exhibited a linear relationship with a good correlation, and the second harmonic of the thermal signal could be used for real-time damage monitoring.

The author reported that a delamination defect affected the phase shift of the thermoelastic temperature change on the surface of SCFRPs, and that the behavior of the phase shift differs depending on the orientation state of the fiber [25]. This phenomenon is considered to be caused by the difference in the thermoelastic modulus of the fiber and resin. The thermoelastic constant *k* of the carbon fiber has a negative value. Therefore, the phase shift of the thermoelastic temperature change occurs, corresponding to the stress sharing condition between the fiber and resin due to the delamination defect. Damage evaluation can also be assessed using the thermoelastic signals and phase shifts of thermal signals on SCFRPs. In this study, TSA was applied to detect fatigue damage in SCFRPs. The effectiveness of the parameters obtained by TSA, such as the thermoelastic temperature change Δ*T*_E_, phase of thermal signal *θ*_E_, and second harmonic temperature component Δ*T*_D_, were investigated. A staircase-like stress level test was performed to investigate the effect of the magnitude of the cyclic loading on the evaluation of the fatigue damage based on these parameters.

## 2. Thermoelastic Stress Analysis Using Infrared Thermography

Dynamic stress changes cause a very small temperature change under adiabatic conditions in a solid. This phenomenon is known as the thermoelastic effect, and is described by Lord Kelvin’s equation, which relates the temperature change (∆*T*_E_) to the changes in the sum of the principal stresses (∆*σ*) under cyclic variable loading as follows:(1)$$\Delta T_{E}=\frac{\alpha }{\rho C_{p}}T\Delta \sigma =-kT\Delta \sigma$$

Here, *α* is the coefficient of thermal expansion, *ρ* is the mass density, *C*_p_ is the specific heat at constant pressure, and *T* is the absolute temperature. The coefficient *k* is called the thermoelastic constant. The change in the sum of the principal stresses (∆*σ*) was obtained by measuring the temperature change (∆*T*_E_) using infrared thermography [9,10,11,12,13].

Because the thermoelastic temperature change is very small and sometimes hidden by the thermal noise of the infrared camera, lock-in infrared thermography using reference signals synchronized with the stress changes is commonly employed to improve the accuracy of stress measurements [26]. The TSA technique uses a lock-in algorithm with a reference-loading signal extracted from the load cell or strain gauge to improve the signal-to-noise ratio.

Because the magnitude of the thermoelastic temperature change was minor, the lock-in thermography method was applied. In this method, a high temperature resolution is realized by performing correlation signal processing with a reference signal, such as a load signal. First, the input reference signal generates the digital data sin(*t*) and cos(*t*) of sine and cosine waves with the same frequency by post-processing at the PC. The correlation coefficient was calculated between the measured temperature data *T*(*t*) and the reference signals sin(*t*) and cos(*t*), respectively, as follows:(2)$$\Delta T_{E,\sin}=\frac{2}{N}\overset{N}{\underset{t=1}{\sum }}T\left(t\right)\sin\left(2\pi t\frac{f_{\mathrm{load}}}{f_{\mathrm{meas}.}}\right)$$
(3)$$\Delta T_{E,\cos}=\frac{2}{N}\overset{N}{\underset{t=1}{\sum }}T\left(t\right)\cos\left(2\pi t\frac{f_{\mathrm{load}}}{f_{\mathrm{meas}.}}\right)$$

Here, *f*_mean_ is the sampling frequency, *f*_load_ is the loading frequency, Δ*T*_E,sin_ and Δ*T*_E,cos_ are the amplitudes of the signal synchronized with the reference signal, respectively, and the amplitude of the signal is synchronized with the opposite phase of the reference signal. In addition, the measurement signal was integrated and averaged to obtain a high-precision amplitude. Because the Δ*T*_E,cos_ component appears when the thermoelastic temperature change deviates from the phase of the load signal, the absolute value Δ*T*_E_ and the phase delay *θ*_E_ of the amplitude of the temperature change can be obtained using the following equation:(4)$$\Delta T_{E}=\sqrt{\Delta T_{E,\sin}{}^{2}+\Delta T_{E,\cos}{}^{2}}$$
(5)$$\theta_{E}=\tan^{-1}\left(-\frac{\Delta T_{E,\cos}}{\Delta T_{E,\sin}}\right)$$

The second harmonic component of the thermal signal is calculated using the following equation: Δ*T*_D,sin_ and Δ*T*_D,cos_ are obtained as the amplitude of the thermal signal *T*(*t*) synchronized with sin(2*t*) and cos(2*t*) with a frequency which is double that of the reference signal [26].
(6)$$\Delta T_{D,\sin}=\frac{2}{N}\overset{N}{\underset{t=1}{\sum }}T\left(t\right)\sin\left(4\pi t\frac{f_{\mathrm{load}}}{f_{\mathrm{meas}.}}\right)$$
(7)$$\Delta T_{D,\cos}=\frac{2}{N}\overset{N}{\underset{t=1}{\sum }}T\left(t\right)\cos\left(4\pi t\frac{f_{\mathrm{load}}}{f_{\mathrm{meas}.}}\right)$$

In metallic materials, the temperature rise due to irreversible energy dissipation occurs at the maximum tensile stress and at the maximum compressive stress during one sinusoidal loading. Therefore, the temperature change due to energy dissipation can be obtained as a component having double the frequency of the load signal. In this study, the magnitude of the second harmonic temperature component *T*_D_ is defined as the temperature change range due to the energy dissipation, and is calculated as follows:

(8)$$T_{D}=2\sqrt{\Delta T_{D,\sin}{}^{2}+\Delta T_{D,\cos}{}^{2}}$$

A schematic illustration of the thermoelastic temperature change is shown in Figure 1. The thermoelastic temperature change in the material with a positive value of *k* shows an opposite phase waveform against the loading waveform. The phase difference was defined as the difference in phase between the thermoelastic temperature change and the loading signal, as shown in Figure 1. For a material with a positive *k* value, the phase difference ∆*θ*_E_ is 180°.

## 3. Experimental Setup

The configurations of the CFRP specimens employed in this study are shown in Figure 2. The specimens were cut from laminated short fiber CFRP sheets with vinyl ester resin and 25.4 mm long carbon fiber bundles. Each bundle was composed of 12,000 short carbon fibers. The mass contents (wt %) of the resin and fiber were 67 and 33, respectively. The specimen had circular notches with a radius of 2 mm.

The fiber orientation angle *φ*_f_ is defined as shown in Figure 3. *φ*_f_ is equal to 0° when the fiber bundle is oriented parallel to the loading axis. The distribution of the fiber orientation angle *φ*_f_ was measured after the TSA measurement. The thin surface layer of the resin was removed by polishing to expose the carbon fiber bundles, and an optical image of the surface carbon fiber was captured using a digital camera. The fiber orientation angle *θ*_f_ was determined using an image processing program developed by Enomae [27].

Cyclic-axis sinusoidal waveform loading with a frequency f of 7 Hz and a stress ratio of *R* = 0.1 was applied to the specimen by an electrohydraulic fatigue testing machine. When the loading frequency is small, thermal diffusion occurs. So accurate thermoelastic stress analysis (TSA) cannot be performed. It was found from preliminary experimental result that the effect of heat diffusion was large at f = 1 Hz, and a loading frequency of 3 Hz or higher was desirable. On the other hand, at f = 9 Hz and above, second harmonic thermal component caused by the fatigue testing machine was observed. Therefore, the load frequency was set to 7 Hz. Microscopic visible images of the specimen surface and side surface were obtained using an optical microscope. The temperature change on the specimen surface was measured by infrared thermography with an MCT array detector (FLIR Systems Inc., Croissy Beaubourg, Fracnce (Wilsonville, OR, USA), SC7500). The specifications and settings of the infrared camera are listed in Table 1.

## 4. Experimental Results

### 4.1. Effect of Damage Generation on Thermoelastic Stress, Phase and Second Harmonic Components

Stress-amplitude constant fatigue tests at *σ*_max_ = 100 MPa, *R* = 0.1 were conducted on smooth specimens (Specimen A). The specimens were broken at *N*_f_ = 40,168 cycles. Figure 3 shows the distribution of the thermoelastic temperature change immediately after the start of the fatigue test and immediately before the fracture, and the infrared image after the rupture, as shown in Figure 3f. In Figure 3f, the white area in the infrared images indicates where the final rupture occurred and the fracture surface appeared. It can be observed from Figure 3 that the area where a high Δ*T*_E_ appears coincides with the fracture area. Figure 4 shows the changes in Δ*T*_E_ and Δ*θ*_E_ in the rectangular area in Figure 3 (Area a-1), where changes in Δ*T*_E_ were observed. Δ*T*_E_ and Δ*θ*_E_ in Figure 4 indicate the average value in the rectangle (range of 5 pixels × 15 pixels). It was found that Δ*θ*_E_ changed from 180° to 0° at *N* = 1500 cycles, and Δ*T*_E_ increased after the change of Δ*θ*_E_ with an increase in the number of cycles. It is considered that the thermoelastic temperature change in the fiber was mainly measured because the fiber orientation angle *θ*_f_ in this area was approximately 0°, and the fiber tended to share stress. Further, and the stress sharing condition between fibers and resin changed owing to the delamination.

Figure 5 shows the result of analyzing the second harmonic temperature component Δ*T*_D_ for the same specimen shown in Figure 3. It was found that the area showing a high Δ*T*_D_ expands with an increase in the number of cycles. The areas showing a high Δ*T*_D_ coincided with the areas where fracture occurred. The load signal and thermal signal at the point showing the high Δ*T*_D_ surrounded by white lines (Area a-2) are shown in Figure 6. As shown in Figure 6, the temperature change is a sinusoidal wave with the opposite phase to the load signal at *N* = 38,000 cycles; whereas the distortion of temperature change, which shows a constant value without a temperature drop, is observed at the maximum tensile load at *N* = 40,000 cycles. Delamination occurred in this evaluation area. Therefore, it is considered that this distortion in the temperature signal is related to the initiation and progression of delamination.

The constant stress amplitude fatigue test for another smooth specimen (Specimen B) was conducted at *σ*_max_ = 180 MPa and *R* = 0.1. This specimen was broken at *N*_f_ = 4.42 × 10^4^ cycles. The distribution of the second-harmonic temperature component is shown in Figure 7. It was found that some regions with high Δ*T*_D_ appeared locally. Figure 8 shows the load and temperature signals at two points (Area b-1 and b-2) in Figure 7. In these regions, the load and temperature signals at *N* = 200 cycles are in phase, and the temperature change in Area b-1 is slightly distorted at the maximum tensile load, compared with that at the minimum tensile load. From Figure 8b, the temperature waveform at *N* = 4000 cycles (life ratio *N/N*_f_ = 0.1) is further distorted than that at *N* = 200 cycles, and no temperature rise occurs above a certain stress magnitude. It was clarified that the distortion of temperature change appears at the maximum tensile load in both cases where the temperature signal is in the opposite phase and in phase with respect to the load signal. The change in the second-harmonic temperature component reflects the distortion of temperature change. It has been reported that the second harmonic component of the thermal signal is caused by the plastic deformation in metallic materials. The distortion of the temperature change and second harmonic temperature component in SCFRPs may be caused by the change in load sharing conditions between the fiber and resin due to delamination damage, the energy dissipation due to the plastic deformation in the resin, and/or the viscoelasticity of resin.

### 4.2. Second Harmonic Temperature Components in Resin Specimens

The constant stress amplitude fatigue tests were conducted resin specimens to investigate the second-harmonic temperature component on the resin. The loading conditions were as follows: maximum stress *σ*_max_ = 32 MPa, stress ratio *R* = 0.1, the repetitive load frequency f = 7 Hz, and the stress waveform was sinusoidal. Fatigue tests were conducted on epoxy and vinyl ester specimens. The shape of specimen is smooth specimen. A strain gauge was set on the surface, opposite to the infrared measurement. The epoxy specimens were broken at *N*_f_ = 2.06 × 10^4^ cycles, and the vinyl ester specimens were broken at *N*_f_ = 6.33 × 10^3^ cycles. Figure 9 shows the change in the second-harmonic temperature component Δ*T*_D_ for the epoxy and vinyl ester specimens. As shown in Figure 9, no change in Δ*T*_D_ was observed during the fatigue test. Figure 10 shows the hysteresis loop of the strain and stress for the vinyl ester specimen. In this stress amplitude, the stress and strain exhibit linear behavior, and the viscoelastic deformation is very small. Because the value of Δ*T*_D_ for epoxy and vinyl ester specimens is smaller than that for SCFRPs, it is considered that the Δ*T*_D_ observed for SCFRP specimens is not due to energy dissipation from viscoelasticity and plastic deformation of the resin.

### 4.3. Relation between the Second Harmonic Temperature Components and Maximum Loading Stress in SCFRP

The distortion of the temperature change occurred at the maximum tension load during cyclic loading. Then, the relationship between the second harmonic temperature component and the magnitude of the maximum stress was investigated. A SCFRP notched specimen (Specimen C) was subjected to staircase-like stress level tests. Figure 11 shows a schematic diagram of the change in the applied maximum stress during the staircase-like stress level test. The specimens were broken at *N*_f_ = 2.57 × 10^4^ cycles. The loading conditions applied were a stress ratio *R* = 0.1, and a load frequency f = 7 Hz.

Figure 12 shows the results of the second harmonic components of the thermal signal and the infrared images at the breaking point, and when loaded with *σ*_max_ = 100 MPa. As shown in Figure 12, the magnitude of Δ*T*_D_ increased with the number of cycles, and the location where a particularly large increase in Δ*T*_D_ was observed coincides with the location where the fracture occurred. The temperature change and the second harmonic temperature component were investigated for each of the following three evaluation areas: Area c-1: no change in Δ*T*_D_ appeared; Area c-2: the high value of Δ*T*_D_ was measured from the beginning but did not change after the test; Area c-3: the location where the fracture occurred and a high value of Δ*T*_D_ was measured during the test.

#### 4.3.1. Area c-1

Figure 13 shows the temperature change in Area c-1, where Δ*T*_D_ did not change. A sinusoidal temperature change was observed regardless of the increase or decrease in the magnitude of the cyclic load. To normalize the magnitude of Δ*T*_D_ during the test, the ratio of Δ*T*_D_ to the thermoelastic temperature change Δ*T*_E_ was calculated (2f/1f). Figure 14 shows the change in 2f/1f during the tests. The change in the average value of 2f/1f across the specimen, and the change in the maximum stress of the cyclic load are plotted in Figure 14. It was found that no change in 2f/1f in Area c-1 was observed, and the magnitude of 2f/1f in Area c-1 was very small as compared to the average value of the whole specimen.

#### 4.3.2. Area c-2

Figure 15 shows the temperature change in Area c-2, where a high Δ*T*_D_ appeared from the beginning of the staircase-like stress level test. As shown in Figure 15, the temperature waveform is distorted at the maximum tension in the initial round of the staircase-like stress level test. When the maximum stress increases, the distortion of the temperature change at the maximum tension increases. The waveform returns to that in the initial round of the fatigue test when the maximum stress decreases. Figure 16 shows the changes in 2f/1f in Area c-2 and the entire surface. The change in 2f/1f is similar to the changes in the maximum stress in the staircase-like stress level test. When the stress amplitude is increased, 2f/1f also increases, and when the stress amplitude is decreased, the value of 2f/1f also returns to its original state. Even when the stress amplitude is increased again, the same value of 2f/1f is shown for the same stress amplitude.

#### 4.3.3. Area c-3

The temperature change and load signal in Area c-3 are shown in Figure 17. As shown in Figure 17, the temperature change at *σ*_max_ = 140 MPa and *N* = 11,200 cycles shows a sinusoidal waveform. On the other hand, the temperature change at *σ*_max_ = 140 MPa and *N* = 12,200 cycles shows a waveform in phase with the load signal, and the temperature rise appears even at the minimum tensile load. When the maximum stress is reduced to 60 MPa, the temperature change returns to the initial waveform. When the maximum stress increased again, the temperature increase at the maximum stress reappeared. The change in 2f/1f in Area c-3 is shown in Figure 18. It was found that the value of 2f/1f in Area c-3 rapidly increased from *N* = 12,200 cycles, and then the 2f/1f changed depending on the maximum tensile stress.

### 4.4. Discussion

The staircase-like stress level tests for SCFRP showed different temperature waveforms in the three assessment areas (Area c-1, Areac-2, and Area c-3).

In Area c-1, a sinusoidal thermal signal was observed during the fatigue test. It is considered that no damage occurred in Area c-1 through the fatigue tests. In Area c-2, a distortion of temperature change was observed at the maximum tension from the early stage of the fatigue test, and the 2f/1f value showed a similar change according to the increase and decrease of the maximum stress in the fatigue test. In Area c-3, no change in 2f/1f was observed in the initial fatigue test, but after *σ*_max_ = 120 MPa, the 2f/1f value changed according to the maximum stress, similar to that in Area c-2. Because Area c-3 coincides with the area where the fracture of the specimen occurred, it is considered that delamination initiated when the change in 2f/1f occurred.

The characteristic temperature change appeared at the maximum tension in both Area c-2 and c-3; and in particular, a temperature rise in phase with the load signal appeared in Area c-3. It is considered that the temperature drop did not occur in Area c-2 because the stress above a certain stress level was not transmitted to the resin due to delamination defects. The thermoelastic constant of carbon fiber has a negative value, so the thermoelastic temperature change of the fiber shows an in-phase change with the load signal. In Area c-3, the thermoelastic temperature change in the fiber was observed mainly due to the stress transfer to the fiber without the stress being shared by the resin caused by the delamination above a certain stress level. This phenomenon is caused by delamination and fiber orientation [25].

The Δ*T*_D_ in Area c-3 was larger than that in Area c-2. Although initial defects might have been present in Area c-2, the final fracture occurred in Area c-3, where a larger Δ*T*_D_ was observed. Therefore, the value of Δ*T*_D_ may correspond to the degree of delamination damage and the propagation rate of damage. When the stress amplitude decreased, it returned to its original value. This phenomenon indicates that the effect of stress sharing condition between the resins and fibers due to delamination was reduced. Under the condition of stress amplitude where the value of Δ*T*_D_ is small, it is expected that the delamination does not propagate, or the propagation rate of the delamination is slow.

The second harmonic temperature component in metals is considered to be due to energy dissipation caused by plastic deformation [28,29], whereas the second harmonic temperature component in SCFRP is considered to be due to the stress sharing condition between the fibers and resins caused by delamination damage. Because the relationship between fatigue damage and the second harmonic temperature component was found, it is considered that the fatigue damage evaluation method using second harmonic components can be applied to damage detection, the evaluation of propagation rate for defect in SCFRP like energy release rate and the further life prediction.

## 5. Conclusions

In this study, the temperature change under cyclic loading was measured for short-fiber-reinforced SCFRP composites. The thermoelastic temperature change Δ*T*_E_, phase of thermal signal *θ*_E_, and second harmonic temperature components Δ*T*_D_ were also evaluated. In the fatigue test of SCFRPs, it was confirmed that changes in Δ*T*_E_, *θ*_E_, and Δ*T*_D_ appeared in the damaged regions. To investigate the generation mechanism of the Δ*T*_D_, fatigue tests were conducted on the resin specimen, and the change in the Δ*T*_D_ was examined. Because the Δ*T*_D_ for the resin specimen was very small compared to that of the SCFRP specimen, it was considered that the Δ*T*_D_ of SCFRP was generated by a different factor from the energy dissipation by the plastic deformation and the viscoelastic deformation of the resin. In addition, a staircase-like stress level test for SCFRPs was carried out, and the relationship between the maximum stress in the cyclic load and the Δ*T*_D_ was examined. A distortion of temperature waveform was observed at the maximum tension, and when the stress level decreased, the temperature waveform returned to the original sinusoidal waveform. Δ*T*_D_ changed according to the change in the maximum stress during the staircase-like stress level test, and a large value of Δ*T*_D_ was observed in the final rupture region. A distortion of temperature change and Δ*T*_D_ were considered to be caused by the change in the stress sharing condition between the fiber and resin due to delamination damage. Therefore, Δ*T*_D_ can be applied to the detection of delamination defects, and the evaluation of damage propagation.

## Figures and Tables

**Figure 1 materials-14-04941-f001:**
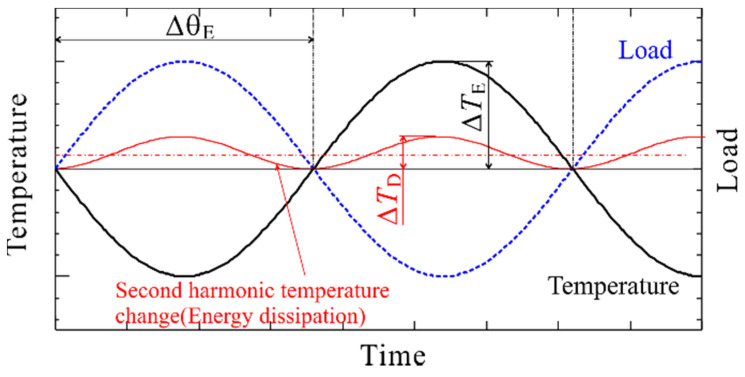
Schematic illustration of thermoelastic temperature change and phase of thermal signal.

**Figure 2 materials-14-04941-f002:**
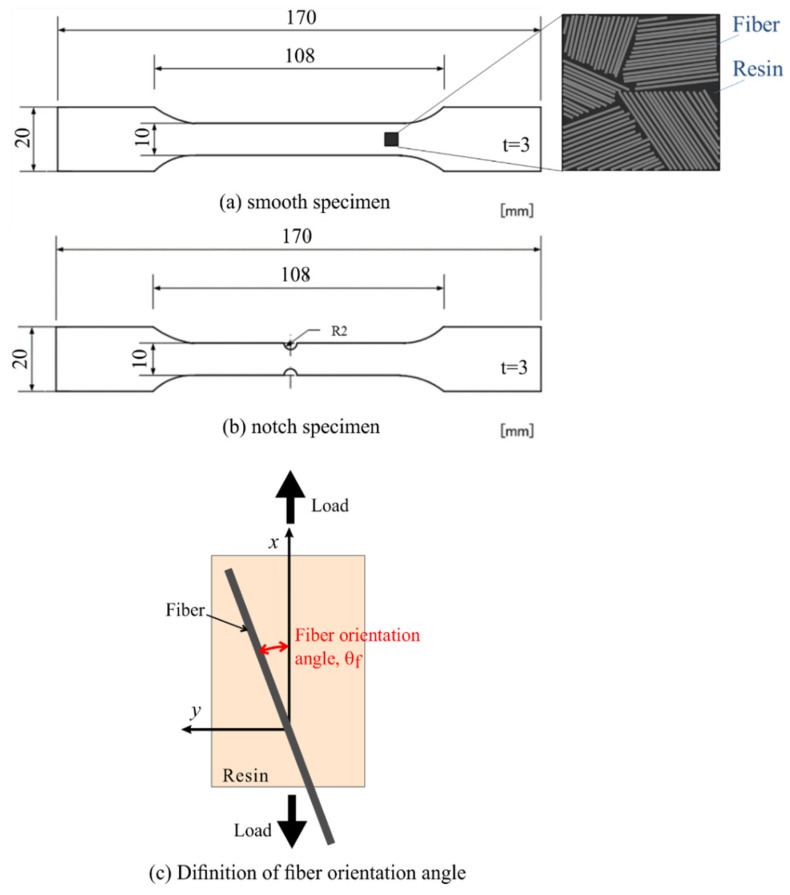
Configurations of the employed short carbon fiber reinforced plastic specimen.

**Figure 3 materials-14-04941-f003:**
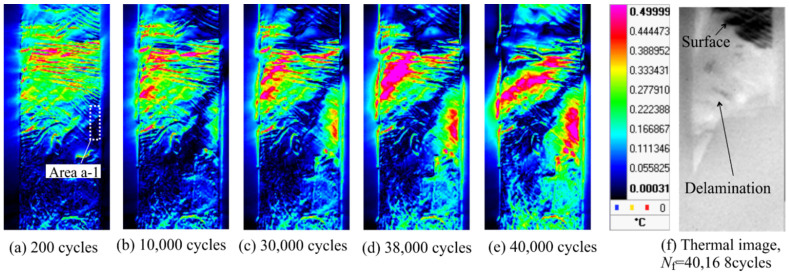
Change in the distribution of thermoelastic temperature change.

**Figure 4 materials-14-04941-f004:**
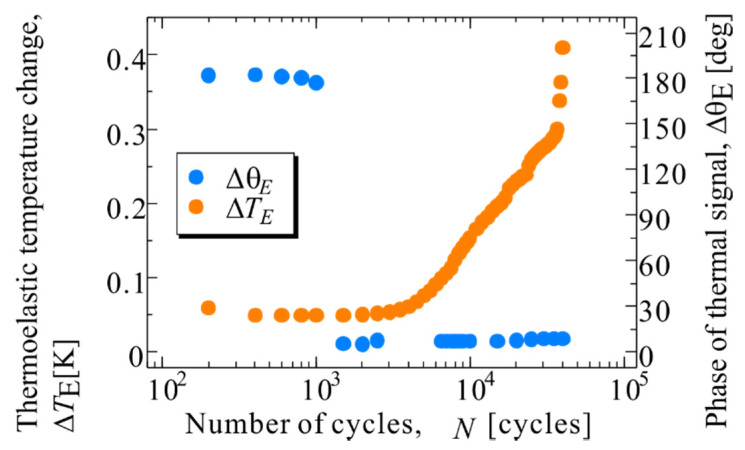
Change in the thermoelastic temperature change and phase of thermal signal in Area a-1.

**Figure 5 materials-14-04941-f005:**
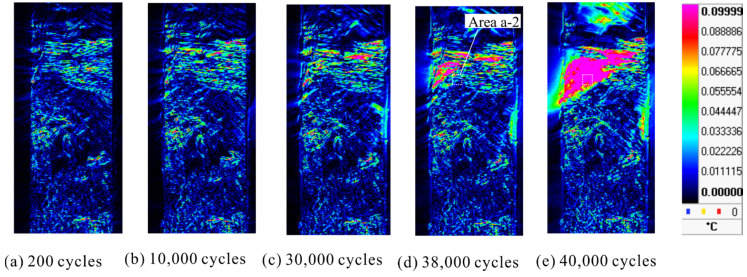
Change in the second harmonic temperature components.

**Figure 6 materials-14-04941-f006:**
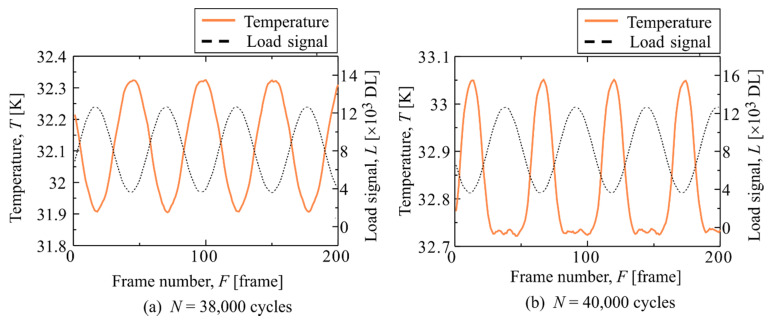
Temperature change and load signal in Area a-2.

**Figure 7 materials-14-04941-f007:**
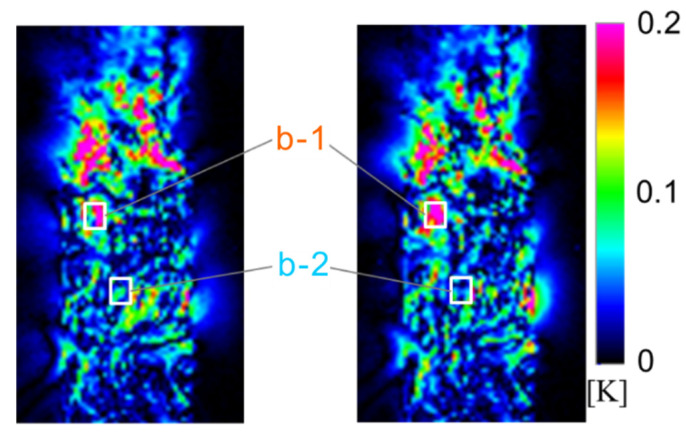
Distribution of second harmonic temperature component for Specimen B.

**Figure 8 materials-14-04941-f008:**
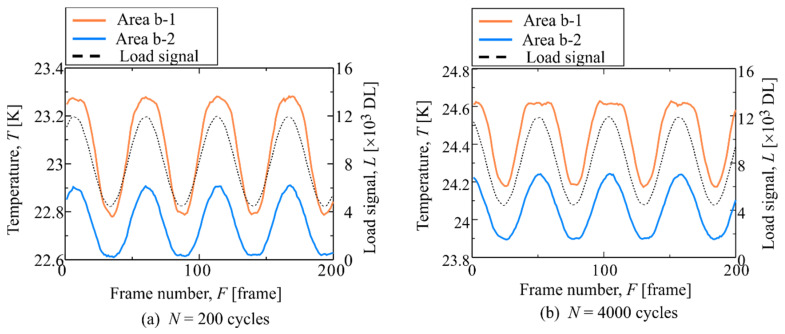
Thermal and load signal at Area b-1 and Area b-2, measured at (**a**) *N* = 200 cycles, (**b**) *N* = 4000 cycles.

**Figure 9 materials-14-04941-f009:**
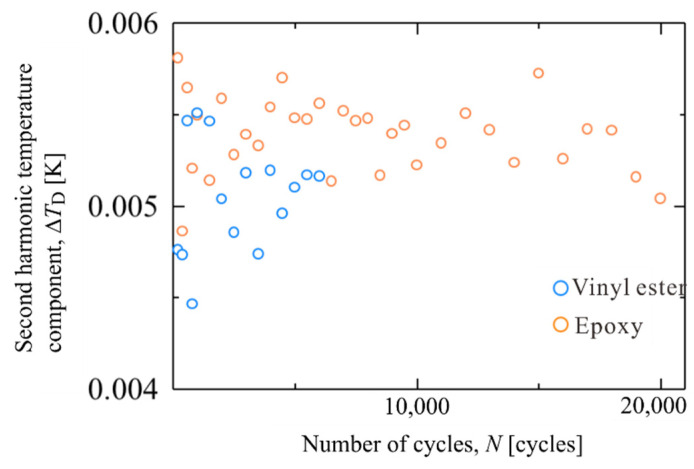
Second harmonic temperature component for epoxy and vinyl ester specimen.

**Figure 10 materials-14-04941-f010:**
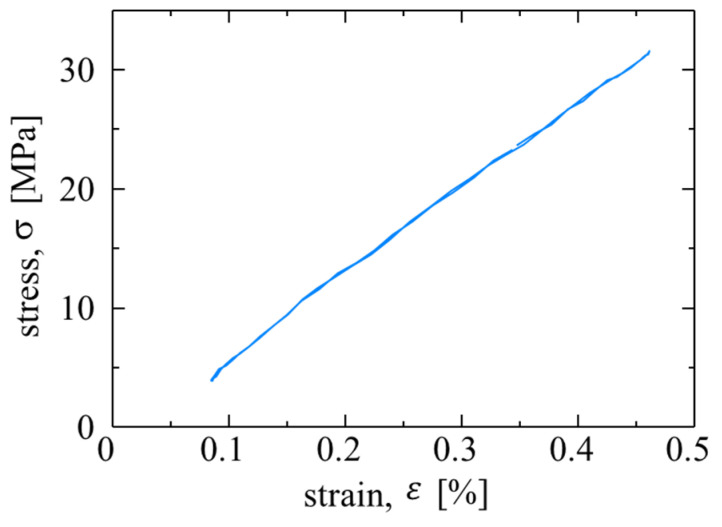
Hysteresis loop of strain and stress for vinyl ester specimen.

**Figure 11 materials-14-04941-f011:**
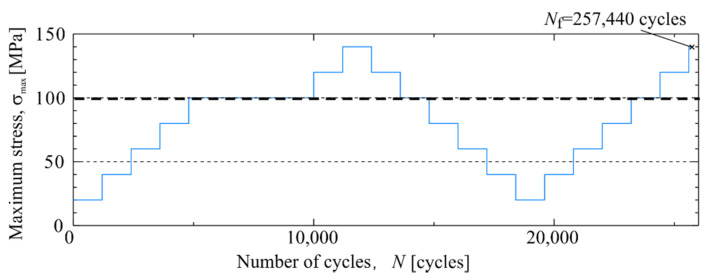
Change of maximum stress applied to the SCFRP specimen during the staircase-like stress level tests.

**Figure 12 materials-14-04941-f012:**
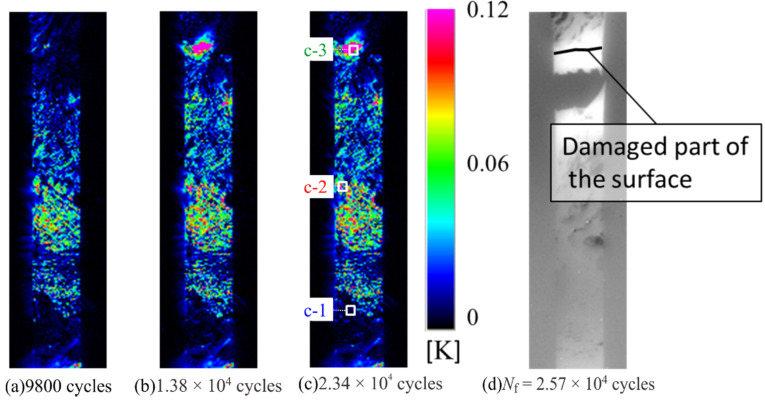
Second harmonic temperature component measured at different timings with *σ*_max_ = 100 MPa, measured at (**a**) *N* = 9800 cycles, (**b**) *N* = 1.38 × 10^4^ cycles, (**c**) 2.34 × 10^4^ cycles, and (**d**) infrared image at *N*_f_ = 2.57 × 10^4^ cycles.

**Figure 13 materials-14-04941-f013:**
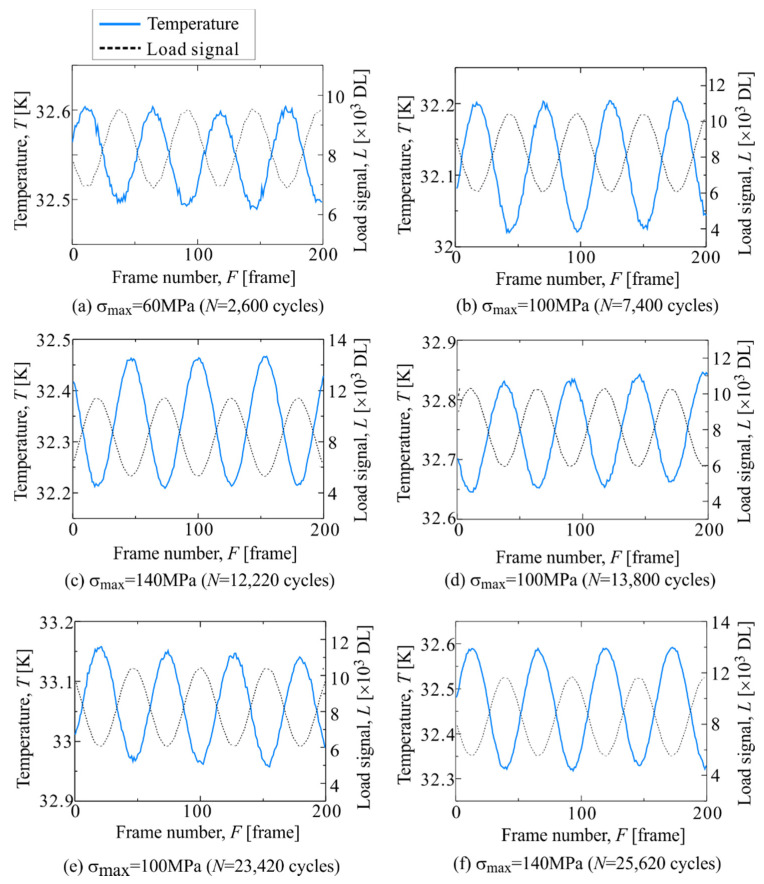
Change in temperature and load signal in Area c-1 measured at (**a**) *σ*_max_ = 60 MPa and *N* = 2600 cycles, (**b**) *σ*_max_ = 100 MPa and *N* = 7400 cycles, (**c**) *σ*_max_ = 140 MPa and *N* = 11,200 cycles, (**d**) *σ*_max_ = 100 MPa and *N* = 13,800 cycles, (**e**) *σ*_max_ = 100 MPa and *N* = 23,420 cycles and (**f**) *σ*_max_ = 140 MPa and *N* = 25,620 cycles.

**Figure 14 materials-14-04941-f014:**
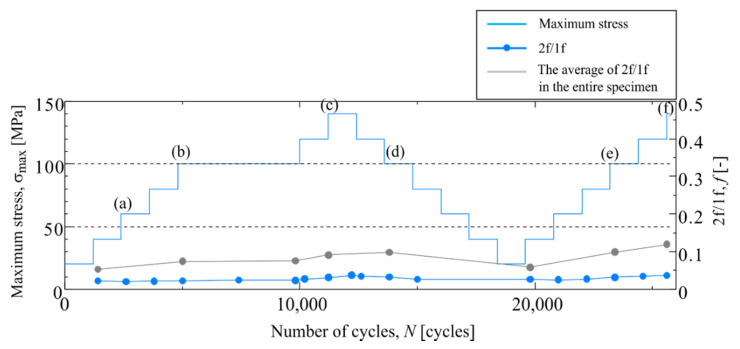
Change in 2f/1f in staircase-like stress level test in Area c-1 The letters (**a**)–(**f**) indicates the measurement timing shown in Figure 13.

**Figure 15 materials-14-04941-f015:**
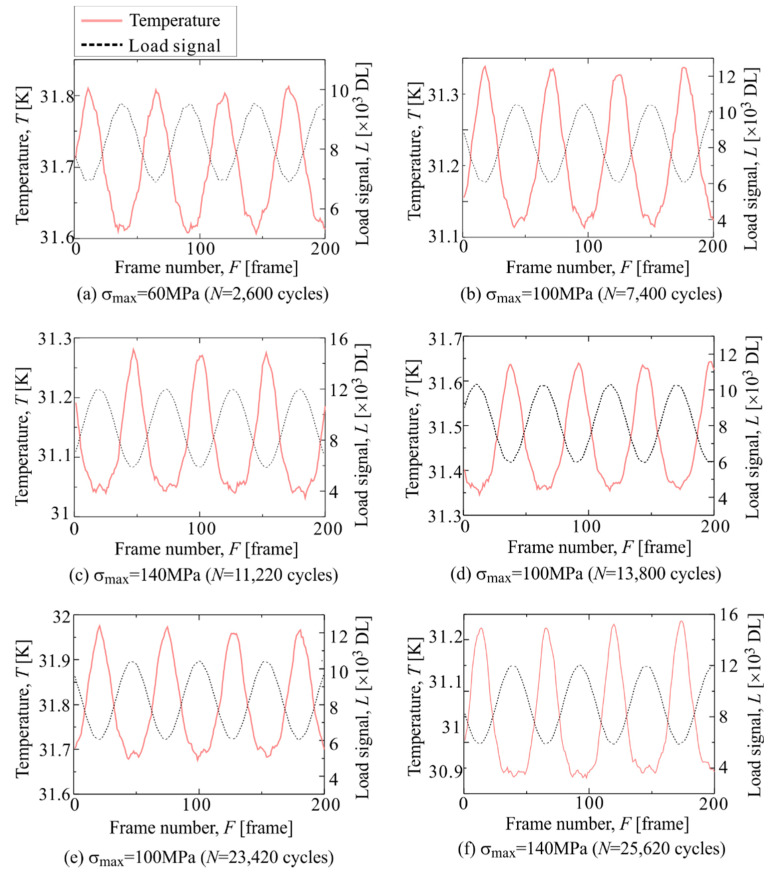
Change in mean temperature and load signal in Area c-2 measured at (**a**) *σ*_max_ = 60 MPa and *N* = 2600 cycles, (**b**) *σ*_max_ = 100 MPa and *N* = 7400 cycles, (**c**) *σ*_max_ = 140 MPa and *N* = 11,200 cycles, (**d**) *σ*_max_ = 100 MPa and *N* = 13,800 cycles, (**e**) *σ*_max_ = 100 MPa and *N* = 23,420 cycles and (**f**) *σ*_max_ = 140 MPa and *N* = 25,620 cycles.

**Figure 16 materials-14-04941-f016:**
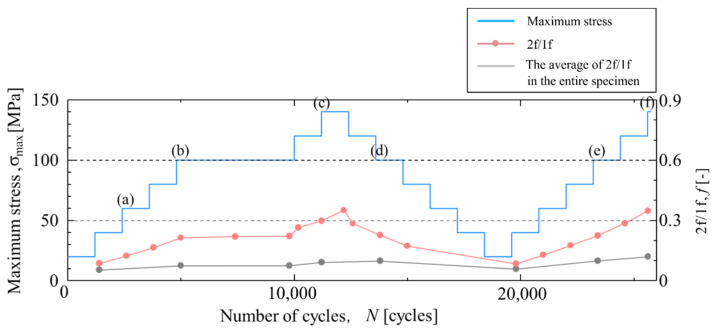
Change in 2f/1f in staircase-like stress level test in Area c-2. The letters (**a**)–(**f**) indicates the measurement timing shown in Figure 15.

**Figure 17 materials-14-04941-f017:**
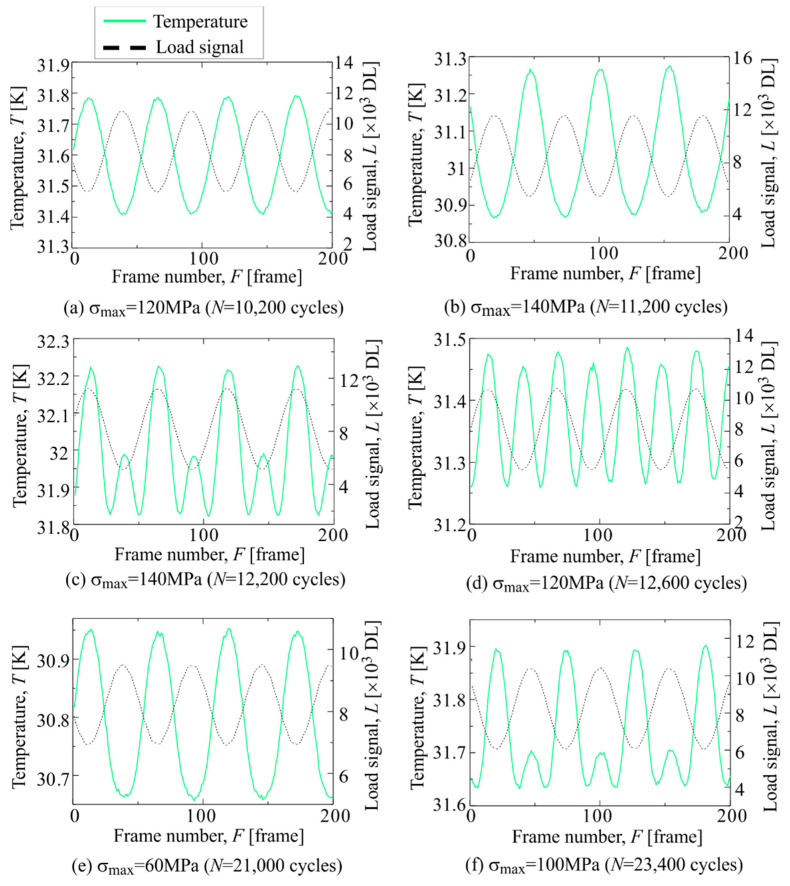
Change in temperature and load signal in Area c-3 measured at (**a**) *σ*_max_ = 120 MPa and *N* = 10,200 cycles, (**b**) *σ*_max_ = 140 MPa and *N* = 11,200 cycles, (**c**) *σ*_max_ = 140 MPa and *N* = 12,200 cycles, (**d**) *σ*_max_ = 120 MPa and *N* = 12,600 cycles, (**e**) *σ*_max_ = 60 MPa and *N* = 21,000 cycles and (**f**) *σ*_max_ = 100 MPa and *N* = 23,400 cycles.

**Figure 18 materials-14-04941-f018:**
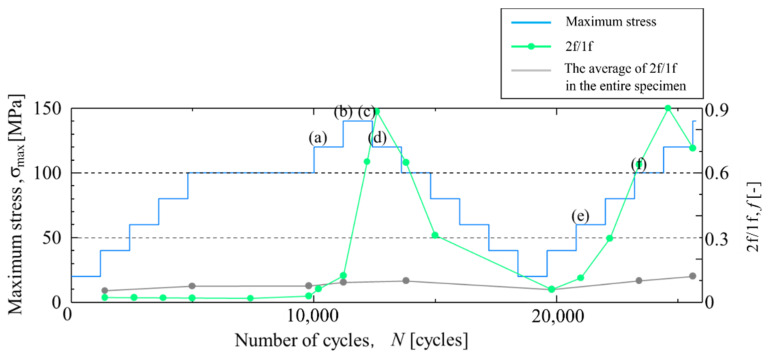
Change in 2f/1f in staircase-like stress level test in Area c-3. The letters (**a**)–(**f**) indicates the measurement timing shown in Figure 17.

**Table 1 materials-14-04941-t001:** Specifications and setting of employed infrared camera.

Infrared Detector	MCT
Detectable wavelength	7.7–9.3 μm
Number of detectors	320 × 256
Temperature resolution (NETD)	25 mK
Framing rate	373 Hz
Time of data acquisition	10 s

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
