# Peer review of "Fatigue Damage Evaluation of Short Carbon Fiber Reinforced Plastics Based on Thermoelastic Temperature Change and Second Harmonic Components of Thermal Signal"

_materials, 2021, doi:10.3390/ma14174941_

Round 1

Reviewer 1 Report

In this work, the authors conducted thermoelastic stress analysis to detect delamination defects in short carbon fiber reinforced plastics, particularly, The thermoelastic temperature change, phase of thermal signal, and second harmonic temperature component were measured and used for damage modeling. The contents of current work should be of interest for the field, however, the following comments should be addressed for further evaluation.

1) For failure analysis of SFRPs, more recent work related to this topic should be strengthened, like “Fragoudakis, R. A numerical approach to determine fiber orientations around geometric discontinuities in designing against failure of GFRP laminates. International Journal of Structural Integrity, 2019, 10(3): 371-379”;

2) For infrared thermography of metallic materials, the theoretical part has been well studied, and references for equations in section 2 should be added;

3) In section 3, for the infrared thermography technique, it relates with the fatigue testing frequency, while the authors conducted fatigue tests under 7Hz, why? How to separate it from the current testing results from those on frequency effects, especially for the CFRP materials? Also the thickness effects of specimens?

4) In Figs. 14, 16 and 18, more discussions on the average 2f/1f curve are needed;

5) Current work has shown good consistency between temperature and load cycles, how about extension for variable amplitude loading condition?

6) It would be better to correlate those thermoelastic temperature parameters with fatigue damage quantificationally, especially for further life prediction. 

Author Response

We wish to thank the reviewer for this comment.  The answers are summarized in a separate file.

Reviewer 2 Report

This is a good paper

Author Response

Thank you for your peer review.

Reviewer 3 Report

Dear Authors,

1) Either material or type of the following specimens is not given. Please, list all specimens used in a table. smooth specimens (Specimen A) [line 153], another specimen (Specimen B) [line 186], Resin Specimens [line 210], A notched SCFRP specimen [line 233]
2) The x-axis labels do not match with the tick marks in Figure 4.
3) Why did the thermoelastic temperature change begin to increase after the phase of thermal signal became nearly 0 deg in Figure 4? Did the same thing happen in different specimens?
4) In Figures 6, 8, 13, 15, and 17, the double y-axis graphs are used. Both ranges of the left and right y-axis are adjusted freely. The waveforms of the temperature and load signal are of similar size. I am afraid that it is rather difficult to see the characteristics of the temperature change. Since the load signal is relatively regular, it is desirable to make it smaller and expose the temperature change. If possible, it is desirable to use a smaller number of different kinds of ranges for the double y-axis. 
5) In Figure 8, the number of cycles is not given.
6) In Figure 14, a smaller range for 2f/1f is used compared to Figures 16 and 18. I am afraid that it may confuse readers.

Author Response

We wish to express our appreciation to the reviewers for their insightful comments on our paper. The comments have helped us significantly improve the paper. 

Please check the attached file containing the answer.

Round 2

Reviewer 1 Report

This work has been well refined, it can be accepted as it is. 

Reviewer 3 Report

Dear Authors,

Thank you for responding to review questions well.